# Virtual Screening of a Library of Naturally Occurring Anthraquinones for Potential Anti-Fouling Agents

**DOI:** 10.3390/molecules28030995

**Published:** 2023-01-19

**Authors:** Gagan Preet, Rishi Vachaspathy Astakala, Jessica Gomez-Banderas, Joy Ebenezer Rajakulendran, Ahlam Haj Hasan, Rainer Ebel, Marcel Jaspars

**Affiliations:** 1Marine Biodiscovery Centre, Department of Chemistry, University of Aberdeen, Aberdeen AB24 3UE, UK; 2The National Decommissioning Centre, University of Aberdeen, Aberdeen AB41 6AA, UK; 3Department of Chemistry, Eastern University, Chenkaladi 30350, Sri Lanka; 4The Medicinal Chemistry and Pharmacognosy Department, College of Pharmacy, Jordan University of Science and Technology, Irbid 22110, Jordan

**Keywords:** in silico, repurposing, pharmacophore, molecular docking, virtual screening, anthraquinones, biofouling, anti-fouling, citreorosein, emodin, paints, marine

## Abstract

Marine biofouling is the undesired accumulation of organic molecules, microorganisms, macroalgae, marine invertebrates, and their by-products on submerged surfaces. It is a serious challenge for marine vessels and the oil, gas, and renewable energy industries, as biofouling can cause economic losses for these industries. Natural products have been an abundant source of therapeutics since the start of civilisation. Their use as novel anti-fouling agents is a promising approach for replacing currently used, harmful anti-fouling agents. Anthraquinones (AQs) have been used for centuries in the food, pharmaceutical, cosmetics, and paint industries. Citreorosein and emodin are typical additives used in the anti-fouling paint industry to help improve the global problem of biofouling. This study is based on our previous study, in which we presented the promising activity of structurally related anthraquinone compounds against biofilm-forming marine bacteria. To help uncover the anti-fouling potential of other AQ-related structures, 2194 compounds from the COCONUT natural products database were analysed. Molecular docking analysis was performed to assess the binding strength of these compounds to the LuxP protein in *Vibrio carchariae*. The LuxP protein is a vital binding protein responsible for the movements of autoinducers within the quorum sensing system; hence, interrupting the process at an early stage could be an effective strategy. Seventy-six AQ structures were found to be highly docked, and eight of these structures were used in structure-based pharmacophore modelling, resulting in six unique pharmacophore features.

## 1. Introduction

Natural products have been a significant source of bioactive compounds since the beginning of the human race. Almost three-quarters of the bioactive compounds present in the market originate from natural sources that are isolated from living organisms, their mimics, or their semisynthetic derivatives [1]. Most commonly, natural compounds are extracted from plants, marine organisms, or microorganism fermentation broths. Amongst these, anthraquinones (AQs) (Figure 1) are some of the most explored natural products [1]. AQs are a class of phenolic compounds characterised by a 9,10-anthracenedione (also called 9,10-dioxoanthracene) core structure substituted by three fused benzene rings with two ketone functional groups on the central ring. In nature, AQs occur (Figure 1) either as glycosides (i.e., linked to a sugar moiety) or as their free aglycones [2]. AQ analogues are a large group of primarily colourful polyketides that exist in either oxidised (anthraquinones) or reduced forms (anthrones, anthranols) or as dimers (dianthrones). Reduction of AQs results in unstable anthrahydroquinones and oxyanthrones [3].

With numerous established bioactivities [4,5,6,7,8,9,10,11,12,13,14,15,16], many natural and synthetic AQs are used in the textile, paint, device, food, cosmetics, and pharmaceutical industries [16,17,18,19,20]. In addition, AQs have been patented for use in building materials to deter destruction by pests, birds, and fungi, highlighting their commercial value [21].

Despite the compounds’ unique characteristics from natural sources, their precise functions, their possible targets and modes of action, and synergies between compounds within complex chemical mixtures remain primarily unknown. Therefore, an integrated approach using technological advances is necessary to successfully develop natural products. These involve the application of efficient selection methods, well-designed extraction/isolation procedures, advanced structure elucidation techniques, and bioassays with high-throughput capacity to establish the bioactivity and patentability of natural products. Several approaches, including molecular modelling, virtual screening of natural product libraries, and database mining, are being used to increase the success of natural product drug discovery research [22].

While AQs have not been extensively explored for their anti-fouling potential, there is ample evidence in the patent literature for their potential suitability to be used as additives in anti-fouling coatings to prevent marine biofouling [23,24].

Marine biofouling is characterised by the undesired accumulation of organic molecules, microorganisms, marine invertebrates, and their by-products on subsea surfaces. This accumulation of marine organisms results in large economic losses for the shipping, oil and gas, and renewable energy industries [25,26]. Another detrimental effect of biofouling on ships’ hulls is the increased emission of greenhouse gases (CO_2_, CO, SO_2_, and NO_2_) into the atmosphere that results from greater fuel consumption needs [27]. Additionally, the colonisation of man-made surfaces facilitates species translocation from one geographical zone to another during the ship’s voyage, either falling off naturally in a new habitat or after cleaning ships’ hulls [28]. The introduction of invasive species into non-native environments can result in competition between species and the eventual eradication of native species. The succession of fouling organisms is generally considered in five main stages, a rapid process which begins with the attachment of bacterial and diatomic cells to a submerged surface (Figure 2), which then leads to the formation of a biofilm. Marine biofilms play an essential role in the subsequent attraction of macroorganism larvae, which are encouraged to settle on these surfaces via various settlement cues [29]. Mussels, barnacles, hard corals, and macroalgae then flourish on these surfaces and, over time, form a dense accumulation of firmly attached organisms which are persistent and therefore very difficult to remove.

To inhibit this process, anti-fouling paints with AF-active additives are used by maritime industries to prevent the attachment of these organisms. Nearly 200 AF-active compounds were identified and isolated from marine microorganisms between 2014 and 2020, indicating that natural products could be used as the inspiration for developing new AF agents [30]. In addition, 112 anti-biofilm, anti-larval, and anti-algal natural products from marine microbes and 26 synthetic analogues were identified from 2000 to 2021 [31]. Examples of naturally potent AF agents include butenolides isolated from a marine *Streptomyces* sp., halogenated compounds from macroalgae, and bromotyrosine-containing compounds from marine sponges [32,33,34,35,36,37,38]. Most commonly, AF activity is expressed through inhibition of microalgal adhesion, inhibition of invertebrate larvae or algal spore settlement, and inhibition of microbial biofilms. When testing for bacterial biofilm inhibitory activity, it is essential to consider that besides exhibiting direct inhibitory activity, compounds of interest may also interrupt the quorum sensing signalling system and thereby inhibit biofilm growth.

Bacteria use cell–cell communication mechanisms to coordinate group behaviours in a cell density-dependent manner, a process known as quorum sensing (QS) [39,40]. Small molecule autoinducers (AIs) are constantly produced and released by bacterial cells [41], and an increase in AIs increases bacterial population density [42,43]. Bacteria monitor the concentration of AIs to track changes in their cell numbers and alter the expressions of a large set of genes in a coordinated manner [44]. These systems not only regulate the expression of genes encoding virulence factors but also other proteins which are involved in primary metabolic processes. A range of 4–10% of the bacterial genome and more than 20% of the bacterial proteome are influenced by QS [45]. Numerous functions, including secretion of virulence factors, competence, sporulation, motility, bioluminescence, antibiotic production, and biofilm formation, are controlled by QS [46]. Therefore, it has been suggested that QS signalling systems are an obvious target for developing novel biofilm growth-inhibiting compounds. 

Gram-negative bacteria use acylated homoserine lactone (AHL) as an AI, while Gram-positive bacteria use autoinducing peptide (AIP) as a means of communication (Figure 3) [47]. In *Vibrio carchariae*, a marine biofilm-forming bacteria involved in the biofouling process, the LuxS protein is the AI-2 synthase in the biosynthetic pathway responsible for the production of AIs [48]. Periplasmic binding protein (LuxP) binds to AI-2 by clamping it between two domains. Then, AI-2-bound LuxP activates the inner membrane protein LuxQ [49]. LuxQ acts as an autophosphorylating kinase at a low cell density that subsequently phosphorylates the cytoplasmic protein LuxU and DNA-binding response regulator LuxO [50]. Phosphorylated LuxO represses the QS response by decreasing production of the known master QS transcription factor LuxR. When AIs enter the periplasmic space at high density and the LuxPQ complex detects them [51], the LuxPQ receptor appears to switch from a kinase state (at low AI-2 concentrations) to a phosphatase state (at high AI-2 concentrations), resulting in the removal of phosphate groups from LuxU. LuxU acts as a kinase and is not able to dephosphorylate LuxO. Finally, AI-1 serves as a species-specific QS signal and regulates LuxO phosphate levels, although through a distinct two-component sensor kinase, LuxO [52,53].

In our previous study, 19 AQ structures related to citreorosein and emodin were tested and reported to show potential anti-fouling (AF) activity against marine biofilm-forming bacteria [54]. These findings prompted us to conduct further research on analogues of these AQ compounds for their AF potential using a computational approach. In the present study, molecular docking analysis (rigid receptor docking (RRD)) was performed to examine the binding mode of 2194 similar AQ compounds into the binding site of LuxP in *V. carchariae* to identify their potential as quorum sensing signalling inhibitors. Based on these results, a pharmacophore study was carried out to generate structure-based pharmacophore features for this study.

## 2. Results and Discussion

### 2.1. Virtual Screening by Molecular Docking Analysis Using a Natural Products Dataset

Based on the bioactivity, SAR, and computational studies mentioned in our previous research, AQs could be possible additives in anti-fouling paints [54]. Therefore, a dataset of 2194 natural AQ-related structures (Appendix A) was taken from the COCONUT database [55] and docked using the same binding site of the LuxP protein of *V. carchariae*, as shown in Figure 4.

The binding affinities range between −11.0 and 0 (kcal/mol). A score for each AQ-related structure was obtained, as shown in Figure 5. However, the plot results show that most AQs have a binding affinity between −10.0 and −6.0 (kcal/mol). Therefore, citreorosein was considered standard in this dataset molecular docking analysis study, with a docking value of −8.2 (kcal/mol), as mentioned in the previous study [25].

Of the 2194 compounds which were screened, 76 compounds were found as top hits with docking values ranging from −8.0 to −10.0 (kcal/mol), as shown in Appendix A. This virtual screening was performed based on the docking score and the presence of natural product details of the compounds in the published data.

### 2.2. Pharmacophore Evaluation (Structure-Based Pharmacophore)

A pharmacophore is an abstract description of the steric and electronic features required to trigger (or block) a biological response. A pharmacophore model can explain how structurally diverse ligands bind at the receptor site based on common interaction points. Structure-based pharmacophore modelling is a commonly used method for developing pharmacophores based on the structural features of the target protein. Structure-based pharmacophore modelling analyses the possible active site in the protein where the interactions of co-crystallised ligands occur, which aids in the design of novel compounds with biological activities of interest. In addition, this method searches for interactions between ligands and the macromolecule. Due to its simplicity, this method is computationally very efficient and is therefore exceptionally well suited for the virtual screening of large compound libraries. From the screening, eight compounds were found with high docking values in the range from −9.8 to −9.0 (kcal/mol): ligands **465D**, **302D**, **52D**, **310D**, **100D**, **298D**, **950D,** and **132D.** Because of their high docking value, these ligands were used for the pharmacophore evaluation (Table 1).

Hydrophobic (**X**), HBA (hydrogen bond acceptor) (****X****), HBD (hydrogen bond donor) (**X**), AR (aromatic) (**X**), PI (positive ionisable area) (**X**), and NI (negative ionisable area) (**X**).

Ligand **465D** interacts with the macromolecule, as shown in Figure 6A,B. It shows hydrophobic interactions with amino acids Ala239A, Phe206A, Trp289A, and Ile211A. This pharmacophore represents a hydrogen bond acceptor (HBA) feature with nearby water molecules and amino acid residue Arg310A. Finally, looking at the hydrogen bond donor (HBD) feature, the ligand interacts with the His180A amino acid residue. Based on the interactions, this structure provides a pharmacophore with three features: hydrophobic interactions (H), hydrogen bond acceptor (HBA), and hydrogen bond donor (HBD).

Ligand **302D** interacts with the macromolecule, as shown in Figure 7A,B. It shows hydrophobic interactions with amino acids Thr266A, Trp82A, Thr134A, Phe206A, Trp289A, and Ile211A. This pharmacophore also has a hydrogen bond acceptor (HBA) feature to nearby water molecules and amino acid residues Trp82A, Ser79A, Arg215A, and Arg310A. Looking at the hydrogen bond donor (HBD) feature, the ligand interacts with Gln77A. Based on the interactions, this structure provides a pharmacophore with three features: hydrophobic interactions (H), hydrogen bond acceptor (HBA), and hydrogen bond donor (HBD).

Ligand **52D** interacts with the macromolecule, as shown in Figure 8A,B. It shows hydrophobic interactions with amino acids Thr266A, Phe206A, and Ala239A. This pharmacophore also has a hydrogen bond acceptor (HBA) feature with nearby water molecules and amino acid residue Arg215A. Looking at the hydrogen bond donor (HBD) feature, the ligand interacts with amino acid residues Pro74A and Asn159A. Based on the interactions, this structure provides a pharmacophore with three features: hydrophobic interactions (H), hydrogen bond acceptor (HBA), and hydrogen bond donor (HBD).

Ligand **310D** interacts with the macromolecule, as shown in Figure 9A,B. It shows hydrophobic interactions with amino acids Thr266A, Tyr81A, Trp82A, Thr134A, and Ile211A. This pharmacophore also has a hydrogen bond acceptor (HBA) feature with nearby water molecules and the Trp82A amino acid residue. Looking at the hydrogen bond donor (HBD) feature, the ligand interacts with amino acid residue SER79A. Based on the interactions, this structure provides a pharmacophore with three features: hydrophobic interactions (H), hydrogen bond acceptor (HBA), and hydrogen bond donor (HBD).

Ligand **100D** interacts with the macromolecule, as shown in Figure 10A,B. It shows hydrophobic interactions with amino acids Phe178A, Tyr81A, Thr266A, Ala239A, and Phe206A. This pharmacophore also has a hydrogen bond acceptor (HBA) feature with nearby water molecules and amino acid residues Asp267A, Asp136A, Thr134A, and Asn159A. Looking at the hydrogen bond donor (HBD) feature, the ligand interacts with the Thr134A amino acid residue. An aromatic (AR) interaction with the Arg310A amino acid residue gives an additional pharmacophoric feature to this structure. Based on the interactions, this structure provides a pharmacophore with four features: hydrophobic interactions (H), hydrogen bond acceptor (HBA), hydrogen bond donor (HBD), and aromatic (AR) interaction.

Ligand **298D** interacts with the macromolecule, as shown in Figure 11A,B. It shows hydrophobic interactions with amino acids Val78A, Thr266A, Val268A, Phe206A, Thr134A, Tyr210A, Ile211A, Phe178A, Trp82A, and Tyr81A. This pharmacophore also has a hydrogen bond acceptor (HBA) feature with nearby water molecules and amino acid residues Ser265A and Val268A. Looking at the hydrogen bond donor (HBD) feature, the ligand only interacts with nearby water molecules. An additional feature, positive ionisable area (PI), shows interactions with Asp136A and Asp267A. Based on the interactions, this structure provides a pharmacophore with four pharmacophore features: hydrophobic interactions (H), hydrogen bond acceptor (HBA), hydrogen bond donor (HBD), and positive ionisable area (PI).

Ligand **950D** interacts with the macromolecule, as shown in Figure 12A,B. It shows hydrophobic interactions with amino acids Tyr81A, Trp82A, Trp289A, Ile211A, and Thr134A. This pharmacophore also has a hydrogen bond acceptor (HBA) feature with nearby water molecules and amino acid residues Thr266A, Arg215A, and Phe206A. An additional feature, negative ionisable area (NI), shows an interaction with Arg310A and Gly288A. Based on the interactions, this structure provides a pharmacophore with three pharmacophore features: hydrophobic interactions (H), hydrogen bond acceptor (HBA), and negative ionisable area (NI).

Ligand **132D** interacts with the macromolecule, as shown in Figure 13A,B. It shows hydrophobic interactions with amino acids Ala239A, Phe206A, Pro109A, and Ile211A. This pharmacophore has hydrogen bond acceptor (HBA) and hydrogen bond donor (HBD) features with nearby water molecules. An aromatic (AR) interaction with Ala239A and Pro109A amino acid residues gives an additional pharmacophoric feature to this structure. Based on the interactions, this structure provides a pharmacophore with four features: hydrophobic interactions (H), hydrogen bond acceptor (HBA), hydrogen bond donor (HBD), and aromatic (AR) interaction.

Based on the above structure-based pharmacophore experiments, there are six possible pharmacophoric features: hydrophobic interactions (H), yellow; hydrogen bond acceptor (HBA), red; hydrogen bond donor (HBD), green, aromatic (AR) interaction, negative ionisable area (NI) interaction, brick red, positive ionisable area (PI) interaction, purple. In the future, these features could make it possible to select and synthesise anthraquinones for formulations (Figure 14).

## 3. Materials and Methods

### 3.1. Origin of Compounds (for Future Anthraquinone Anti-Fouling Study)

To identify potential structures for future anti-fouling research, the anthraquinone dataset of 2194 structures was taken from the natural products database COCONUT (collection of open natural products) [55].

Molecular docking analysis was performed using Autodock Vina v.1.2.0 (The Scripps Research Institute, La Jolla, CA, USA) docking software [56,57]. The receptor site was predicted using the MOE Site Finder program [58], which uses a geometric approach to calculate putative binding sites in a protein, starting from its tridimensional structure. This method is not based on energy models but on alpha spheres, which are a generalisation of convex hulls [59]. The X-ray crystal structure of LuxP in complex with AI-2 (PDB: 1JX6) was retrieved from the Protein Data Bank and utilised to perform docking simulations [60]. The box centre and size coordinates were −18.2833 × −9.13497 × 22.3052 and 18.4165 × 9.67157 × 39.6987 around the active site. All coordinates used Angstrom units. Default search parameters were used when the number of binding modes was 10, the exhaustiveness was 8, and the maximum energy difference was 3 kcal/mol. Samson by OneAngstrom, 2022, [61], Chimera 1.16 [62], and LigPlot+ software [63] were used to visualise and calculate protein–ligand interactions.

### 3.2. 3D Pharmacophore Model Generation (Structure-Based)

LigandScout 4.4.8 (Inte: Ligand, Vienna, Austria) advanced software [64] was used to generate the 3D pharmacophore models. LigandScout’s algorithm calculates and displays chemical interactions between protein–ligand complexes.

## 4. Conclusions

In this computational study, based on the results shown in our previously published study, we report 76 AQ-based structures which may have anti-fouling capacity when used as additives in anti-fouling paints. In silico screening of these structures displayed exciting interactions with the binding site of the previously reported protein and exhibited good binding potentials. The best eight docked AQ structures were used for structure-based pharmacophore modelling experiments, which revealed six unique pharmacophore features that may help guide future studies in identifying, selecting, and designing anthraquinone structures from the large compound library or synthetically producing those that could work individually or synergistically with other additives for anti-fouling purposes. The authors also hope that such studies will help select AQ structures that are readily available and cost-effective in the future.

## Figures and Tables

**Figure 1 molecules-28-00995-f001:**
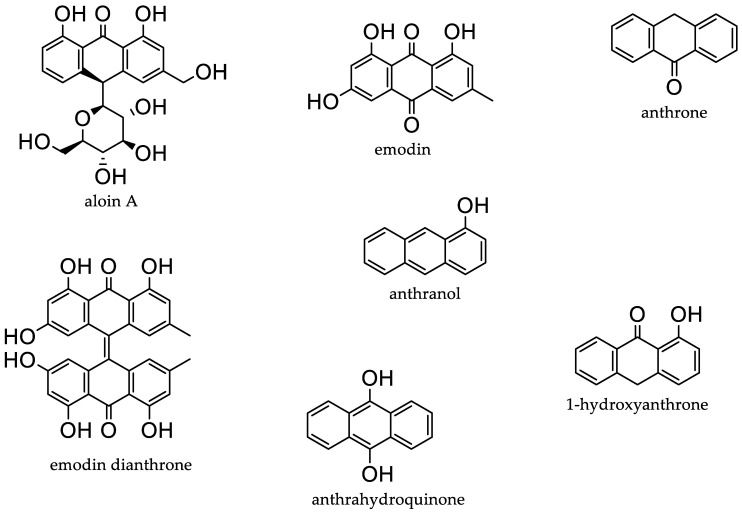
Examples of different anthraquinones.

**Figure 2 molecules-28-00995-f002:**
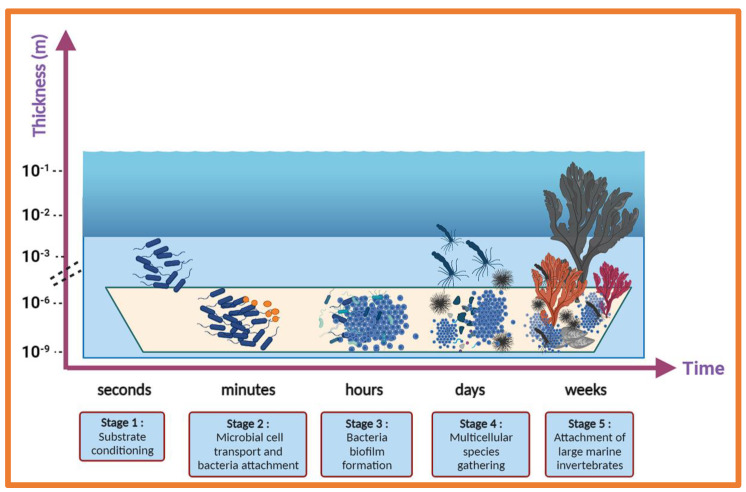
Five stages of typical marine biofouling: (1) substrate conditioning, (2) transport of microbial cells to the surface and attachment of bacteria on the surface, (3) formation of a microbial biofilm, (4) development of a complex community, and (5) attachment of larger marine invertebrates.

**Figure 3 molecules-28-00995-f003:**
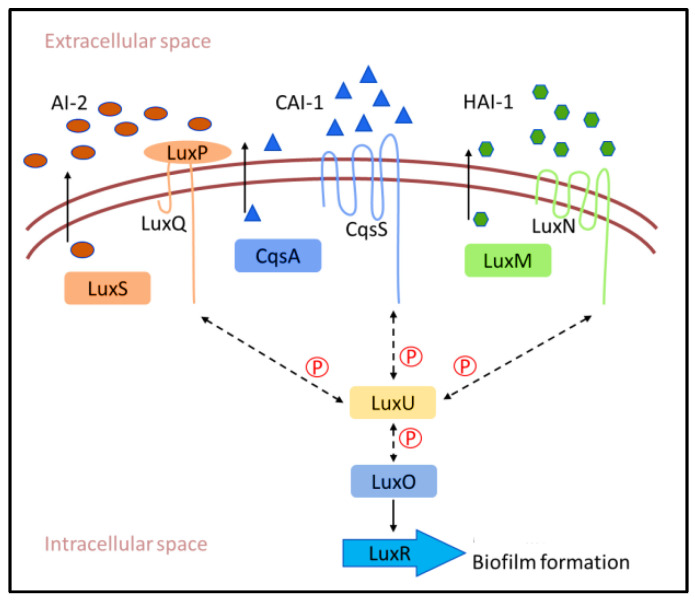
The quorum sensing mechanism of *V. carchariae*.

**Figure 4 molecules-28-00995-f004:**
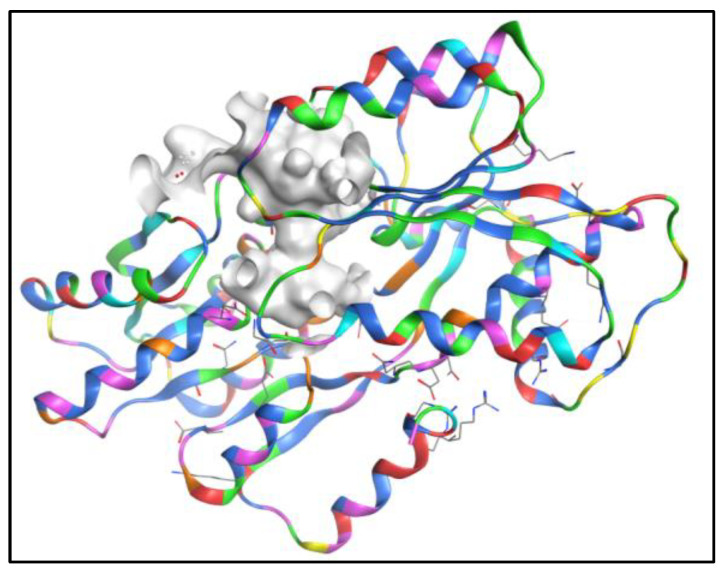
The docking site (grey/white colour) of the LuxP protein in *V. carchariae*.

**Figure 5 molecules-28-00995-f005:**
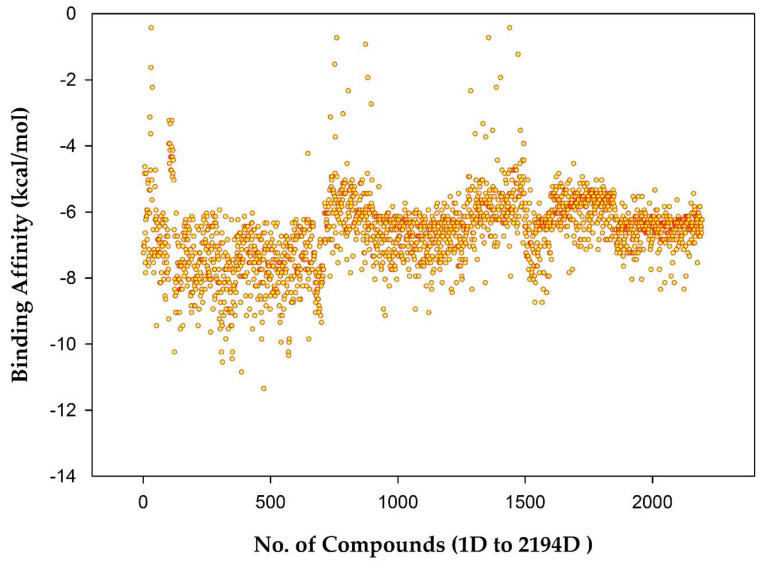
Plot showing Binding Affinity (kcal/mol) vs. No. of compounds (1D to 2194D).

**Figure 6 molecules-28-00995-f006:**
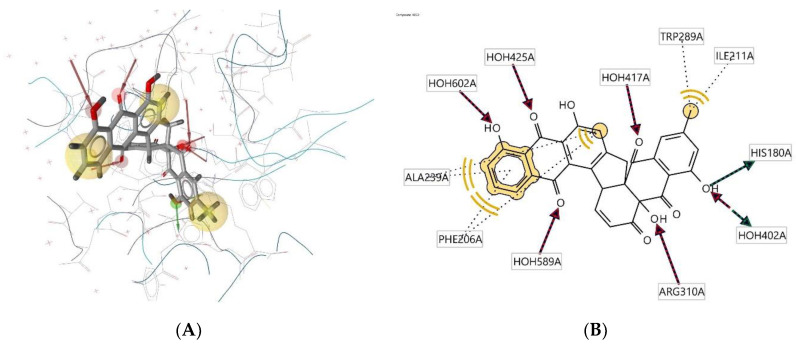
Ligand **465D** interacts with the macromolecule. (**A**) 3D view of pharmacophore at macromolecule binding site. (**B**) 2D view of the pharmacophore. Pharmacophore features: hydrophobic interactions (H), yellow; hydrogen bond acceptor (HBA), red; hydrogen bond donor (HBD), green.

**Figure 7 molecules-28-00995-f007:**
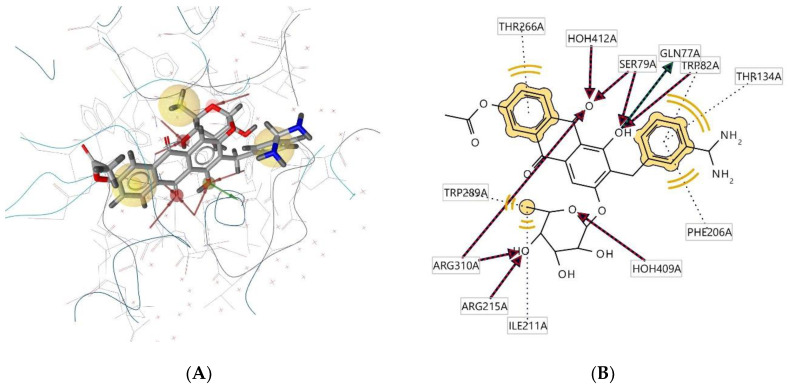
Ligand **302D** interacts with the macromolecule. (**A**) 3D view of pharmacophore at the macromolecule binding site. (**B**) 2D view of pharmacophore. Pharmacophore features: hydrophobic interactions (H), yellow; hydrogen bond acceptor (HBA), red; hydrogen bond donor (HBD), green.

**Figure 8 molecules-28-00995-f008:**
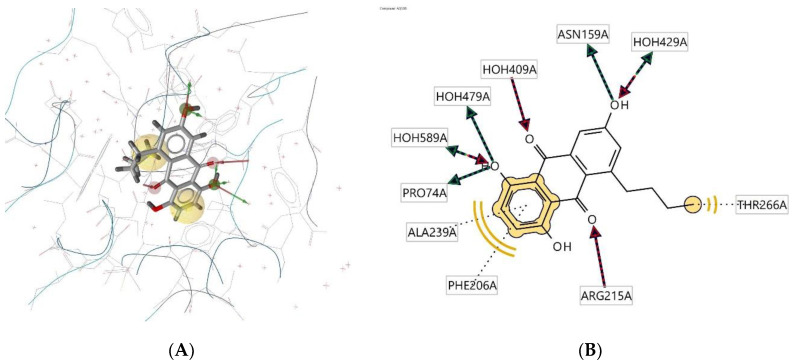
Ligand **52D** interacts with the macromolecule. (**A**) 3D view of pharmacophore at the macromolecule binding site. (**B**) 2D view of pharmacophore. Pharmacophore features: hydrophobic interactions (H), yellow; hydrogen bond acceptor (HBA), red; hydrogen bond donor (HBD), green.

**Figure 9 molecules-28-00995-f009:**
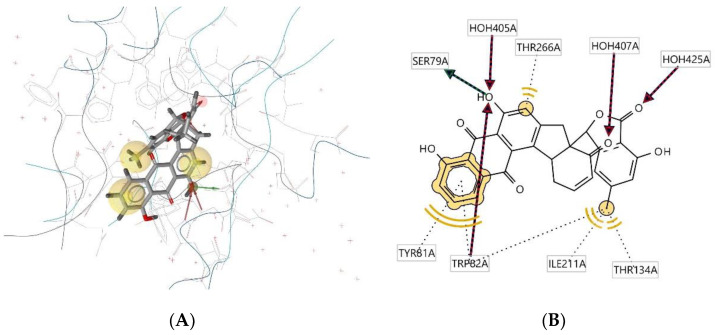
Ligand **310D** interacts with the macromolecule. (**A**) 3D view of pharmacophore at the macromolecule binding site. (**B**) 2D view of pharmacophore. Pharmacophore features: hydrophobic interactions (H), yellow; hydrogen bond acceptor (HBA), red; hydrogen bond donor (HBD), green.

**Figure 10 molecules-28-00995-f010:**
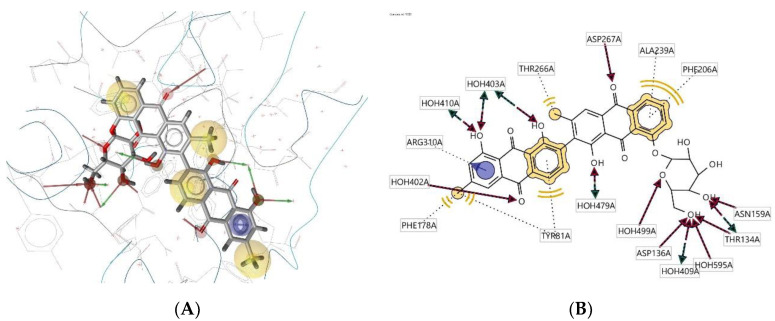
Ligand **100D** interacts with the macromolecule. (**A**) 3D view of pharmacophore at the macromolecule binding site. (**B**) 2D view of pharmacophore. Pharmacophore features: hydrophobic interactions (H), yellow; hydrogen bond acceptor (HBA), red; hydrogen bond donor (HBD), green; aromatic (AR) interaction, purple.

**Figure 11 molecules-28-00995-f011:**
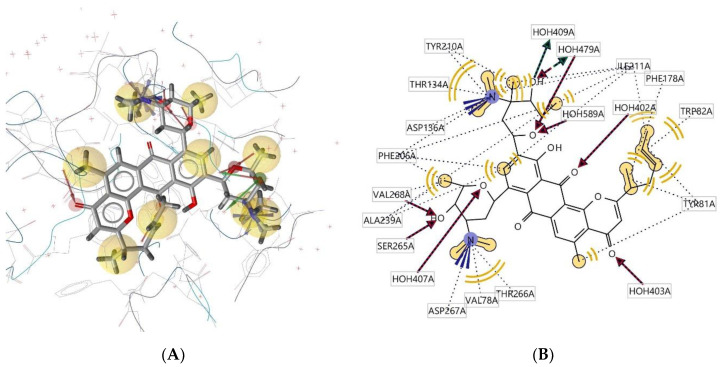
Ligand **298D** interacts with the macromolecule. (**A**) 3D view of pharmacophore at the macromolecule binding site. (**B**) 2D view of pharmacophore. Pharmacophore features: hydrophobic interactions (H), yellow; hydrogen bond acceptor (HBA), red; hydrogen bond donor (HBD), green; positive ionisable area (PI) interaction, purple.

**Figure 12 molecules-28-00995-f012:**
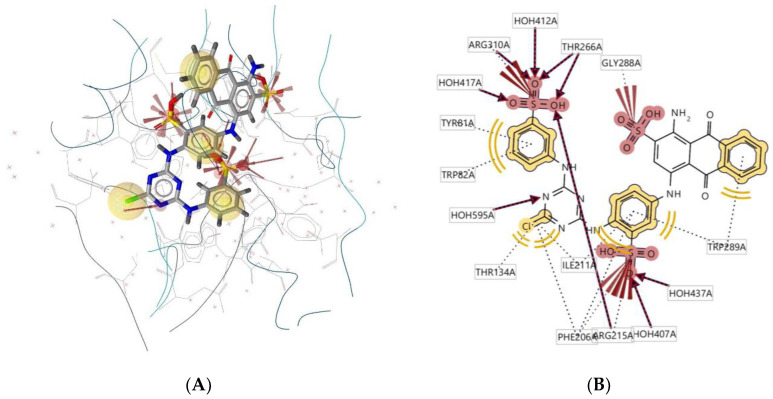
Ligand **950D** interacts with the macromolecule. (**A**) 3D view of pharmacophore at the macromolecule binding site. (**B**) 2D view of pharmacophore. Pharmacophore features: hydrophobic interactions (H), yellow; hydrogen bond acceptor (HBA), red; negative ionisable area (NI) interaction, brick red.

**Figure 13 molecules-28-00995-f013:**
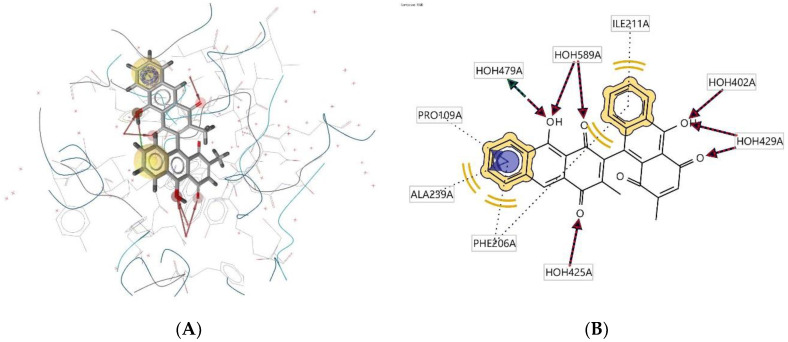
Ligand **132D** interacts with the macromolecule. (**A**) 3D view of pharmacophore at the macromolecule binding site. (**B**) 2D view of pharmacophore. Pharmacophore features: hydrophobic interactions (H), yellow; hydrogen bond acceptor (HBA), red; hydrogen bond donor (HBD), green; aromatic (AR) interaction, purple.

**Figure 14 molecules-28-00995-f014:**
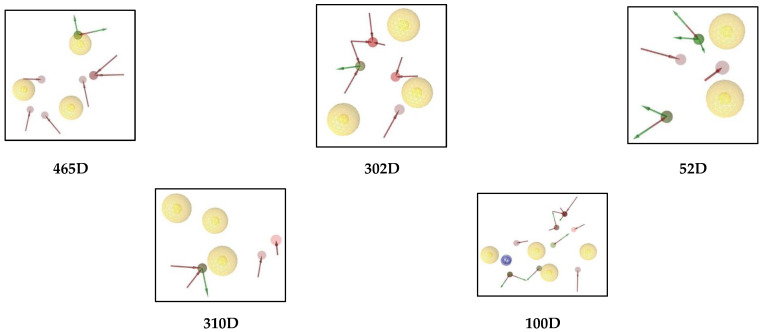
Six unique pharmacophore features: hydrophobic interactions (H), yellow; hydrogen bond acceptor (HBA), red; hydrogen bond donor (HBD), green; aromatic (AR) interaction; negative ionisable area (NI) interaction, brick red; positive ionisable area (PI) interaction, purple.

**Table 1 molecules-28-00995-t001:** Table showing eight ligands and types of interactions with nearby residues.

Residues	465D	302D	52D	310D	100D	298D	950D	132D
Ala239A	**X**		**X**		**X**			**X** **X**
Arg215A		**X**	**X**				**X**	
Arg310A	**X**	**X**			**X**		**X**	
Asn159A			**X**		**X**			
Asp136A					**X**	**X**		
Asp267A					**X**	**X**		
Gln77A		**X**						
Gly288A							**X**	
His180A	**X**							
Ile211A	**X**	**X**		**X**		**X**	**X**	**X**
Phe178A					**X**	**X**		
Phe206A	**X**	**X**	**X**		**X**	**X**	**X**	**X**
Pro74A			**X**					
Pro109A								**X** **X**
Ser79A		**X**		**X**				
Ser265A						**X**		
Thr134A		**X**		**X**	**X** **X**	**X**	**X**	
Thr266A		**X**	**X**	**X**	**X**	**X**	**X**	
Trp82A		**X** **X**		**X** **X**		**X**	**X**	
Trp289A	**X**	**X**					**X**	
Tyr81A				**X**	**X**	**X**	**X**	
Tyr210A						**X**		
Val78A						**X**		
Val268A						**X** **X**		

## Data Availability

Not applicable.

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
