# Peer review of "Virtual Screening of a Library of Naturally Occurring Anthraquinones for Potential Anti-Fouling Agents"

_molecules, 2023, doi:10.3390/molecules28030995_

Round 1
Reviewer 1 Report
This is a good piece of work, similar to the authors previous work (10.3389/fntpr.2022.990822). The conducted research is interesting, the results are well presented and the final conclusions are valid.
There are just a few minor points:
Is it possible to make a higher resolution image on page 1? This one is murky, but it could be due to the pdf conversion.
Instead of „molecular docking experiment“ the authors should use „molecular docking analysis“ or simply „molecular docking“.
Line 157
Is it correct to write “(-8.2 kcal/mol)”? I see that you usually put only the measuring unit in the brackets, hence this question.
Under Table 1., meaning of the abbreviations used should be stated. I know they are explained later in the text, but abbreviations should be explained at the part of the text where they are first mentioned.
Author Response
"Please see the attachment."

Reviewer 2 Report
The paper by Preet et al. presents a computation study for screening AQs-based compounds by molecular docking and pharmacophore modeling. The result is okay as per the screening and pharmacophore modeling. The manuscript is very clear and the main conclusions are supported by the computational results. The topic is of interest. I suggest acceptance with revision.
The work lacks experimental support to make a substantial statement on the compound’s predicted binding affinity. Performing docking and pharmacophore modeling is good, however, there should be some experimental validation in that connection.
I could not understand the logic of the selection of the LuxP protein of V. carchariae. Is there any other target we can use?
Could you find some of the common features in the 76 compounds?
I would also suggest using FEP-based calculation or other docking methods for the remaining 76 structures to confirm the strength of binding affinity. And performing a MD based simulation will also reveals structural stability and molecular interactions.
Author Response
"Please see the attachment."

Reviewer 3 Report
Virtual screening of a library of naturally occurring anthraquinones for potential antifouling agents
Gagan Preet et al.
The paper by Gagan Preet et al. used virtual screening and modeling studies to identify the potential antifouling agents from 2194 compounds, mostly anthraquinones or related structures taken from the coconut NP database. This work identified 76 leads and resulted in further studies by pharmacophore modeling to 6 best compounds.
The paper emphasized the fouling and biofilm formation of microbes on subsea surfaces, its implication for the oil, gas, and energy industries, and the importance of natural compounds as antifouling agents rather than chemicals in preventing and mitigation efforts will result in decarbonization and less pollution.
The authors have highlighted the three ways the compounds of interest may interrupt fouling by the bacterial biofilm inhibitory activity, quorum sensing signaling inhibitory system, and inhibiting biofilm growth.
In the present study, authors used Rigid Receptor Docking (RRD) to understand the binding mode of 2194 similar AQ compounds into the binding site of LuxP in Vibrio carchariae to identify their potential as quorum-sensing signaling inhibitors
The author's methodology is thorough, and their justification is at length. There is enough literature survey and supporting literature to their discussion. The authors provided enough conclusions.
Minor Comments
1. The abstract does mention LuxP protein in V. carchariae. A one-liner is required for the significance of considering this protein
Author Response
"Please see the attachment."

Round 2
Reviewer 2 Report
I am satisfied. Should be accepted in my opinion.